# Changes in SARS-CoV-2 viral load and mortality during the initial wave of the pandemic in New York City

**Michael J. Satlin**[1,2]☯*, **Jason Zucker**[3]☯, **Benjamin R. Baer**[4]☯, **Mangala Rajan**[5], **Nathaniel Hupert**[6,7], **Luis M. Schang**[8], **Laura C. Pinheiro**[5], **Yanhan Shen**[9,10], **Magdalena E. Sobieszczyk**[3], **Lars F. Westblade**[1,2], **Parag Goyal**[5,11], **Martin T. Wells**[4,12], **Jorge L. Sepulveda**[13], **Monika M. Safford**[5]

1 Division of Infectious Diseases, Department of Medicine, Weill Cornell Medicine, New York, New York, United States of America, 2 Department of Pathology and Laboratory Medicine, Weill Cornell Medicine, New York, New York, United States of America, 3 Division of Infectious Diseases, Department of Medicine, Columbia University Irving Medical Center, New York, New York, United States of America, 4 Department of Statistics and Data Science, Cornell University, Ithaca, New York, United States of America, 5 Division of General Internal Medicine, Department of Medicine, Weill Cornell Medicine, New York, New York, United States of America, 6 Department of Population Health Sciences, Weill Cornell Medicine, New York, New York, United States of America, 7 Cornell Institute for Disease and Disaster Preparedness, New York, New York, United States of America, 8 College of Veterinary Medicine, Cornell University, Ithaca, New York, United States of America, 9 Gertrude H. Sergievsky Center, Vagelos College of Physicians and Surgeons, Columbia University Irving Medical Center, New York, New York, United States of America, 10 Department of Epidemiology and Biostatistics, CUNY Graduate School of Public Health and Health Policy, New York, New York, United States of America, 11 Division of Cardiology, Department of Medicine, Weill Cornell Medicine, New York, New York, United States of America, 12 Division of Biostatistics and Epidemiology, Weill Cornell Medicine, New York, New York, United States of America, 13 Department of Pathology, George Washington University, Washington, DC, United States of America

☯ These authors contributed equally to this work.

* mjs9012@med.cornell.edu

**Data Availability Statement:** All relevant data are within the manuscript and its Supporting Information files.

## Abstract

Public health interventions such as social distancing and mask wearing decrease the incidence of severe acute respiratory syndrome coronavirus 2 (SARS-CoV-2) infection, but it is unclear whether they decrease the viral load of infected patients and whether changes in viral load impact mortality from coronavirus disease 2019 (COVID-19). We evaluated 6923 patients with COVID-19 at six New York City hospitals from March 15-May 14, 2020, corresponding with the implementation of public health interventions in March. We assessed changes in cycle threshold ($C_T$) values from reverse transcription-polymerase chain reaction tests and in-hospital mortality and modeled the impact of viral load on mortality. Mean $C_T$ values increased between March and May, with the proportion of patients with high viral load decreasing from 47.7% to 7.8%. In-hospital mortality increased from 14.9% in March to 28.4% in early April, and then decreased to 8.7% by May. Patients with high viral loads had increased mortality compared to those with low viral loads (adjusted odds ratio 2.34). If viral load had not declined, an estimated 69 additional deaths would have occurred (5.8% higher mortality). SARS-CoV-2 viral load steadily declined among hospitalized patients in the setting of public health interventions, and this correlated with decreases in mortality.

**Funding:** This work was partially supported by the National Center for Advancing Translational Sciences of the National Institutes of Health (UL1 TR0023484 to Julianne Imperato-McGinley) and the National Institute of Allergy and Infectious Diseases (UM1 AI069470 to M.E.S). The funders had no role in study design, data collection and analysis, decision to publish, or preparation of the manuscript.

**Competing interests:** I have read the journal's policy and the authors of this manuscript have the following competing interests: L.F.W reports receiving consulting and grant support from Roche Molecular Systems. M.M.S receives grant support from Amgen, Inc. All other authors declare no competing interests.

## Introduction

Severe acute respiratory syndrome coronavirus 2 (SARS-CoV-2) has caused a public health crisis, leading to more than 175 million infections and nearly 4 million deaths from coronavirus disease 2019 (COVID-19) as of June 20, 2021 [1]. Early in the pandemic, patients hospitalized with COVID-19 had reported mortality rates of 20–25% [2–4]. We previously demonstrated that SARS-CoV-2 viral load on admission, as measured by cycle threshold ($C_T$) values from reverse transcription-polymerase chain reaction (RT-PCR) assays, is independently associated with the risk of in-hospital mortality among patients hospitalized with COVID-19 [5–7]. These findings have also been observed by groups at other medical centers [8, 9].

In New York City (NYC), progressively more stringent non-pharmacological public health interventions, such as restrictions on large gatherings, school closures, closure of all non-essential businesses, and recommendations to wear masks when out of the household, were implemented between March 12-April 2, 2020 [10–12]. Although imposition of these measures was associated with a subsequent decline in the daily incidence of COVID-19 cases in NYC after April 5 [13], we do not know whether these measures led to a decrease in SARS-CoV-2 viral load among infected patients. Furthermore, we do not know the degree to which changes in viral load among infected patients may have led to changes in outcomes among hospitalized patients. Therefore, we conducted an observational study at six hospitals in NYC with the following objectives: 1) to assess changes in SARS-CoV-2 viral load among patients presenting to the emergency department (ED) with COVID-19 during the first two months of the COVID-19 pandemic; 2) to assess changes in in-hospital mortality among hospitalized patients during this time; 3) to explore how changes in viral load relate to changes in in-hospital mortality.

## Materials and methods

### Study population and setting

This is an observational study of patients with symptomatic COVID-19 who presented to the EDs at NewYork-Presbyterian Hospital (NYP)/Weill Cornell Medical Center, NYP Lower Manhattan Hospital, NYP Queens, NYP/Columbia University Irving Medical Center, NYP Allen Hospital, or NYP Morgan Stanley Children's Hospital between March 15, 2020 and May 14, 2020. These hospitals range from academic quaternary care centers to community hospitals, span three NYC boroughs, and serve a diverse patient population. This diverse group of hospitals was selected in order to maximize the generalizability of our findings. Patients were included in the study if they had a positive SARS-CoV-2 RT-PCR test using the cobas SARS-CoV-2 assay (Roche Molecular Systems, Inc., Branchburg, NJ) or Xpert Xpress SARS-CoV-2 assay (Cepheid, Inc., Sunnyvale, CA) within one day of ED presentation. March 15, 2020 corresponds to the first day that SARS-CoV-2 RT-PCR testing was performed at NYP clinical microbiology laboratories. Patients who were transferred to a different hospital and for whom we were unable to ascertain whether they were discharged alive were excluded. Only a patient's first ED presentation with COVID-19 that met inclusion criteria was analyzed. Institutional Review Board approvals for the study were obtained from Weill Cornell Medicine, Columbia University Irving Medical Center, and NewYork-Presbyterian Queens, with a waiver of informed consent.

For three of the six hospitals, all data were abstracted manually from the electronic medical record and entered into a REDCap database for the entire study period, as previously reported [6, 14]. The same data entry protocol for the other three hospitals was followed for patients presenting from March 15, 2020 to April 6, 2020. After April 6, data from these three hospitals

were electronically abstracted from a clinical data warehouse without manual review. For each patient, age and demographics, comorbidities, hospital of presentation, admission to the hospital, and use of hydroxychloroquine, remdesivir, corticosteroids, and interleukin-6 inhibitor therapies, were recorded. Duration of COVID-19-related symptoms prior to ED presentation, obesity, and race/ethnicity data were available for a subset of patients.

## SARS-CoV-2 viral load assessment

The cobas and Xpert Xpress SARS-CoV-2 RT-PCR assays were performed for routine clinical care according to the manufacturer's instructions [15, 16]. The cobas assay was initiated on March 15, 2020, and the Xpert Xpress assay was initiated on different dates in April 2020 at each hospital. The viral load assessment was based on $C_T$ values for the ORF1ab gene using the cobas assay and $C_T$ values for the N2 gene using the Xpert Xpress assay. Both gene targets are specific for SARS-CoV-2. We converted these $C_T$ values into qualitative assessments of viral load using definitions that have been shown to predict in-hospital mortality [5, 6]. For the cobas assay, high, medium, and low viral loads were defined as $C_T$ value <25, $C_T$ value 25–30, and $C_T$ value >30, respectively. For the Xpert Xpress assay, high, medium, and low viral loads were defined as $C_T$ value <27, $C_T$ value 27–32, and $C_T$ value >32, respectively. The differences in viral load cutoffs are based on a median two cycle difference between assays [17].

## Statistical analyses

We first characterized mean $C_T$ values for each day of the study, including 95% prediction intervals, with results stratified by the type of SARS-CoV-2 RT-PCR assay. $C_T$ values were modeled using gamma regression with the day of ED presentation and assay type as covariates. Given differences in the upper range of $C_T$ values for the cobas and Xpert Xpress assays, any positive tests that had a $C_T$ value >40 were assigned a value of 40 cycles for this analysis [6]. For each day, we also determined the proportion of patients who presented with high, medium, and low viral loads, using assay-specific definitions outlined above. We then constructed a log-link binomial generalized additive model of presenting with a high viral load by day of ED presentation, adjusting for age, gender, comorbidities, and type of RT-PCR assay [18]. All terms other than day of ED presentation were taken to be linear and the smoothing parameter for day of ED presentation was selected via generalized cross validation [19]. The first day of the study was used as a baseline and this adjusted risk ratio was plotted over time. We chose this model because the log-link allows estimation of risk ratios rather than odds ratios and because using a generalized additive model rather than a generalized linear model allows the relationship between the day of ED presentation and the log risk ratios to be non-linear. Duration of symptoms was not included in the primary model because this data point was only available for approximately one-half of study subjects, but an additional model that included duration of symptoms was constructed as a sensitivity analysis.

Next, we plotted the proportion of hospitalized patients with COVID-19 who died during their hospitalization based on their day of ED presentation, with 95% Wilson Score intervals [20]. We then constructed a multivariable logistic regression model with in-hospital mortality as the dependent variable. Variables included in this model were age, gender, number of patients admitted to the hospital with COVID-19 on the same day (an indicator of hospital strain), comorbidities at presentation, hospital of admission, use of remdesivir, steroids, and IL-6 inhibitors, and admission viral load category (high, medium, or low). The predicted number of deaths and proportion of hospitalized patients who died based on day of ED presentation using this model were then compared to the actual number and proportions of deaths. We calculated the predicted number of deaths by aggregating the predicted probabilities. The

Hosmer-Lemeshow test (with an across-time variant) was used to assess calibration and how well this model fit to the actual observed events [21].

In order to assess the potential impact of changes in viral load on mortality, we estimated the number of additional patients who would have died had every patient presented with a viral load that followed the probabilistic distribution on the first day of the study, yet all other attributes remained constant. To do this, we first used the gamma regression of $C_T$ values to construct a counterfactual viral load for each patient, where the expected $C_T$ values for each assay were kept constant throughout the study, using the expected $C_T$ values from the first day of the study. We then used the multivariable logistic regression model to estimate the counter-factual predicted mortality probability for each patient using the counterfactual viral load. Next, we plotted the counterfactual predicted number of deaths for each day of ED presenta-tion by aggregating the counterfactual predicted mortality probabilities. Finally, we compared the number of predicted deaths that would have occurred had the distribution of viral load not changed from the first day of the study to the number of predicted deaths using the viral loads that were actually observed in the study. Note, since the logistic regression has an intercept, the sum of predicted probabilities (using the viral loads that were actually observed) is equal to the number of observed deaths in the study. This difference in number of deaths was calculated together with a 95% bootstrap confidence interval (CI) based on model uncertainties [22].

## Results

### Study population

There were 7595 patients who presented to the ED with COVID-19 at the six study hospitals during the study period, of whom 6923 were eligible for the study (S1 Fig). Overall, the median patient age was 63 years (interquartile range [IQR] 50–76; S1 Table). The median age increased from 58 years during the first week of the study, peaked at 68 years at the fifth week of the study, then declined to 59 years during the last week of the study. Forty-five percent of patients were women and 76% of patients with data on race or ethnicity were Black or of Hispanic eth-nicity. Hypertension, diabetes, chronic pulmonary disease, and coronary artery disease were present in 39%, 27%, 14%, and 12% of patients, respectively. Based on data that were available for approximately half of the patients, the median duration of symptoms prior to ED presenta-tion was 6 days (IQR: 3–9). Seventy-one percent of patients who presented to the ED were admitted to the hospital. COVID-19 therapies included hydroxychloroquine (64% of patients), remdesivir (4%), corticosteroids (20%), and IL-6 inhibitors (5%).

### Changes in viral load over time

Mean $C_T$ values of SARS-CoV-2 RT-PCR tests increased throughout the study period for each assay, indicating lower viral loads over time (Fig 1). For the cobas assay, the mean $C_T$ value during the first week of the study was 25.4 and the mean $C_T$ value during the last week of the study was 30.9. For the Xpert Xpress assay, the mean $C_T$ value during the first week of its implementation (third week of study) was 31.0 and the mean $C_T$ value during the last week the study was 37.0. The proportion of patients presenting with a high viral load also declined throughout the course of the study (Fig 2). In the first week of the study, 47.7% of patients pre-sented with a high viral load, compared to only 7.8% of patients during the last week of the study. In a multivariable model that adjusted for age, gender, comorbidities, and type of RT-PCR assay, the adjusted risk of presenting with a high viral load declined throughout the study period (S2 Fig). Similar risks were observed in a sensitivity analysis that included dura-tion of symptoms in the multivariable model (S3 Fig).

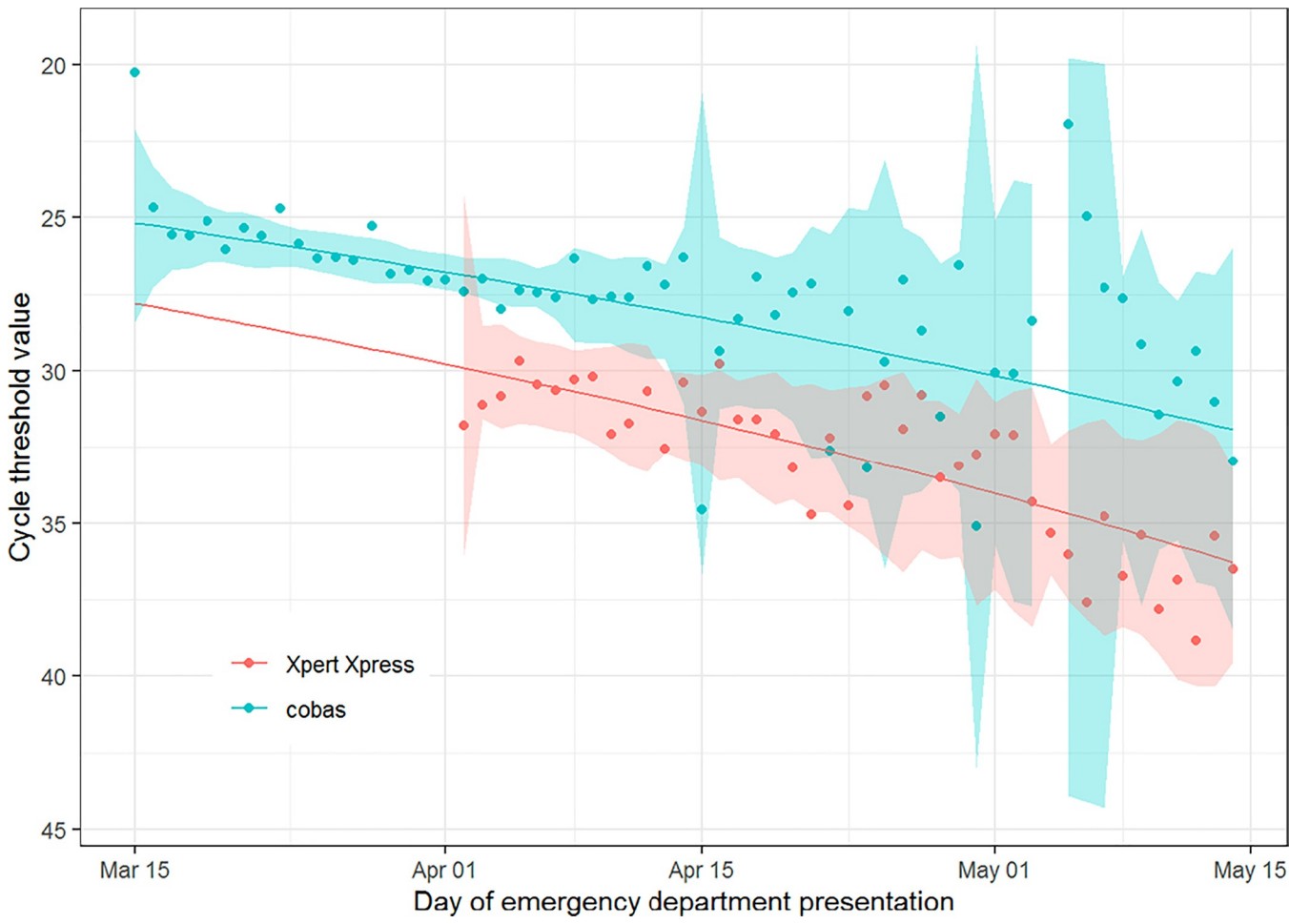

**Fig 1. Cycle threshold values from SARS-CoV-2-specific RT-PCR tests among patients presenting to the emergency department with COVID-19 over time, separated by type of SARS-CoV-2 assay.** Shaded areas represent 95% prediction intervals and lines are best-fit curves from a Gamma regression model. The ratio of expected $C_T$ values for each assay type on consecutive days was 1.004 (95% CI: 1.0036–1.0046).

### Changes in in-hospital mortality over time

The proportion of hospitalized patients with COVID-19 who died in the hospital based on day of ED presentation are displayed in Fig 3. Overall, 1183 (24.2%) of 4887 hospitalized patients died during their hospitalization. Unlike the steady decline observed for viral load, in-hospital mortality first increased from 14.9% during the first study week to 28.4% during the third study week, and then steadily declined to 8.7% during the last study week.

### Impact of changes in viral load on mortality

In a multivariable logistic regression model, presenting with a medium viral load (adjusted odds ratio [aOR] 1.51; 95% CI: 1.24–1.83) or high viral load (aOR 2.34; 95% CI: 1.96–2.80) was associated with increased in-hospital mortality compared to a low viral load (Table 1). The predicted numbers of patients who died and proportions of patients who died based on this multivariable model are similar to the actual numbers and proportions of deaths that were observed (Figs 4 and 5: red vs. black lines). The number of deaths in the counterfactual predictive model, where admission viral load remains constant from the first study day instead of declining, began to separate from the model using observed viral loads in the second week of the

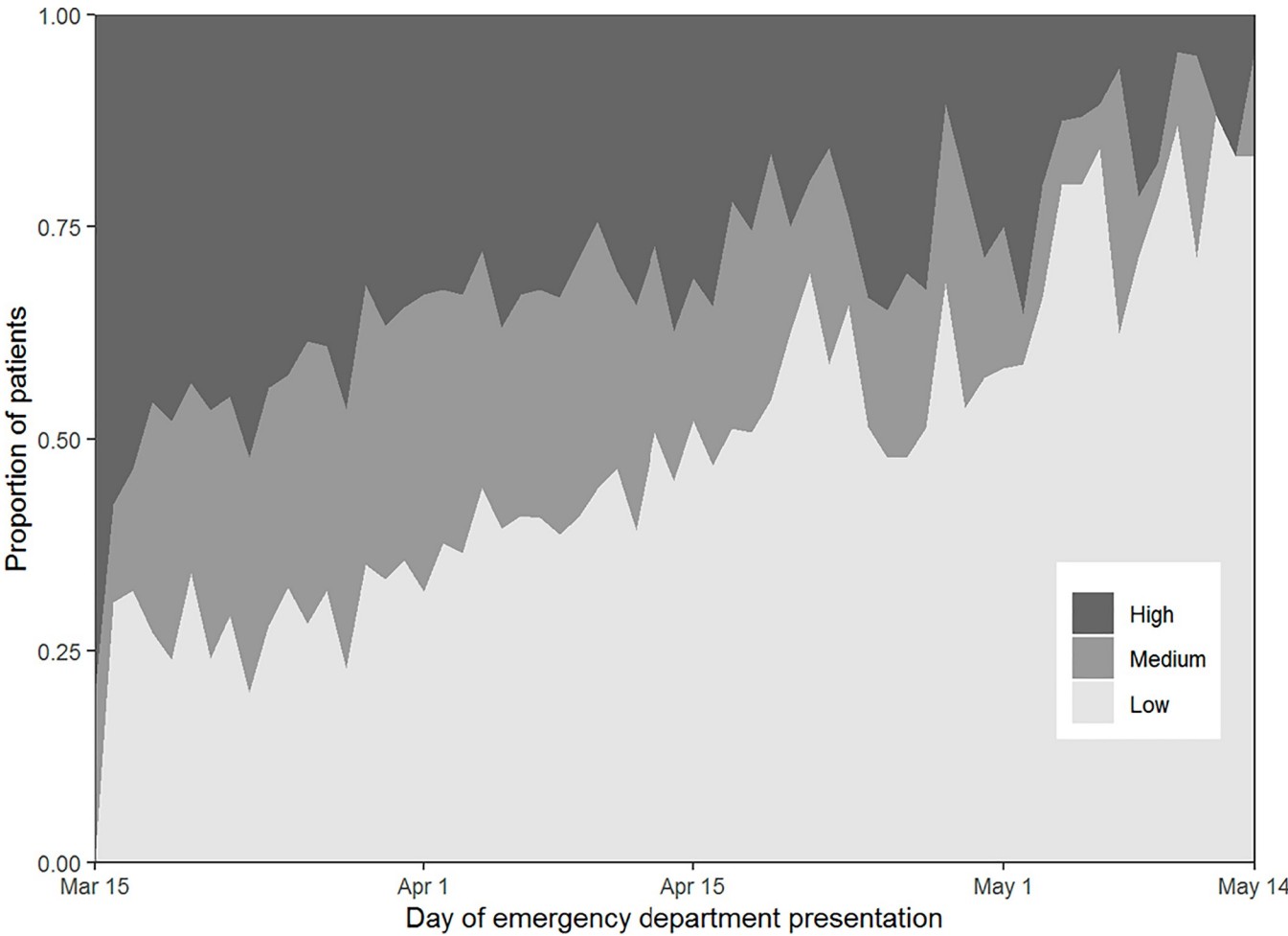

**Fig 2. Proportions of patients presenting to the emergency department with high, medium, and low SARS-CoV-2 viral loads over time.**

study (Figs 4 and 5: blue lines vs. red lines). This model predicts that if viral load had stayed constant and not declined, there would have been 69 additional deaths (95% CI: 50–90) throughout the study period, or a 5.8% greater number of deaths (95% CI: 4.2–7.6%).

## Discussion

This study demonstrates that SARS-CoV-2 viral load at the time of presentation to the hospital steadily declined throughout the first two months of the pandemic in the setting of implementing non-pharmacological public health interventions to reduce interpersonal interactions and aerosol exposures in NYC. In-hospital mortality among patients with COVID-19 followed a slightly different pattern: it initially increased until the peak of the pandemic was reached in early April, followed by a steady decline. Admission viral load was an independent predictor of in-hospital mortality and counterfactual modeling suggests that if viral load had not declined over the study period, an additional 5.8% of patients would have died compared to the number of deaths actually observed.

The steady decline in admission SARS-CoV-2 viral load during the first wave of the pandemic in NYC mirrors the experience reported from an Italian hospital during their first wave of the COVID-19 pandemic [23]. We found that viral load decreased starting on the first study

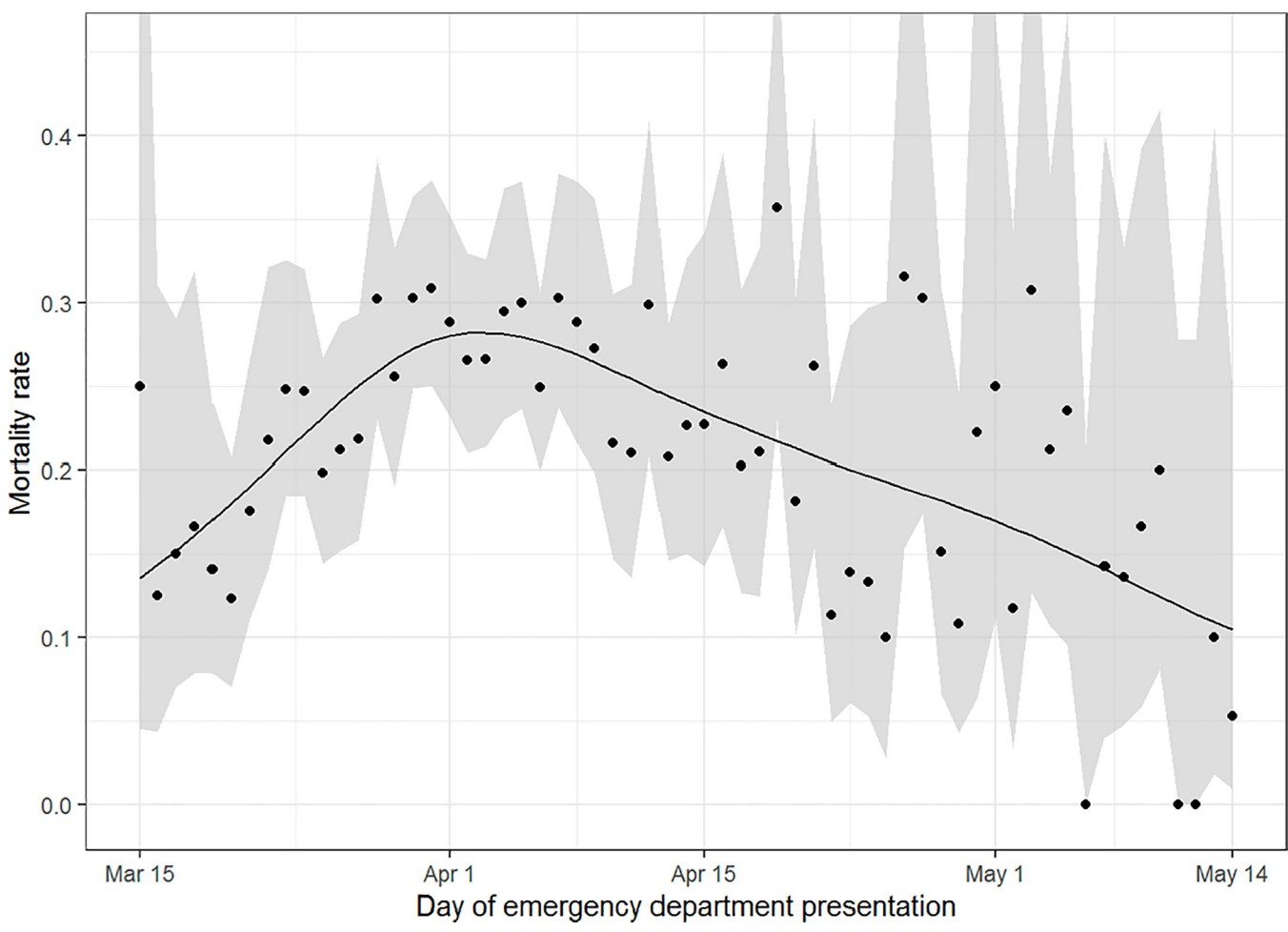

**Fig 3. Proportion of hospitalized patients with COVID-19 who died in the hospital over time.** Davies' test for logistic regression shows that the increase then decrease in the mortality proportion over time is statistically significant ($P<10^{-10}$).

day (March 15, 2020) and this decrease occurred 2–3 weeks prior to the decrease in the number of new cases in NYC [13]. This finding suggests that changes in viral load could be a leading indicator to future changes in the incidence of COVID-19 infections and thus could be a potential public health tool to predict the evolution of the pandemic. For example, if this hypothesis is correct, monitoring viral loads could help guide public health agencies in deciding on the timing of opening and closing venues such as schools and restaurants. We encourage further research to assess the impact of viral load monitoring in different settings and waves of the pandemics.

The reasons for the decline in SARS-CoV-2 viral load during the first wave of the pandemic in NYC warrant further investigation. It is unlikely that the decline is related to changes in testing practices given that declines in viral load were observed among the subset of patients who were hospitalized, and all hospitalized patients received SARS-CoV-2 RT-PCR tests during the study period. Furthermore, declines in viral load were observed after adjusting for potential confounding factors, including duration of symptoms (S3 Fig), and were observed using both RT-PCR testing platforms (Fig 1). We hypothesize that one or more of the non-pharmacological public health interventions instituted by New York State and City governments to decrease interpersonal contact and aerosol exposures may have contributed to the decline in viral loads.

**Table 1. Logistic regression model of factors associated with in-hospital mortality among patients admitted with COVID-19.**

| Characteristic | Adjusted odds ratio (95% CI) | P |
|---|---|---|
| Age, per year | 1.06 (1.06–1.07) | <0.001 |
| Female gender | 0.66 (0.57–0.77) | <0.001 |
| Number of patients admitted on same day, per 10 patients | 1.03 (1.02–1.03) | <0.001 |
| Comorbidities | | |
| Hypertension | 1.04 (0.85–1.28) | 0.7 |
| Diabetes mellitus | 1.27 (1.05–1.54) | 0.013 |
| Chronic pulmonary disease | 0.91 (0.74–1.10) | 0.4 |
| Coronary artery disease | 1.11 (0.91–1.34) | 0.3 |
| Treatments | | |
| Hydroxychloroquine | 1.03 (0.86–1.22) | 0.8 |
| Remdesivir | 1.35 (0.92–1.96) | 0.12 |
| Corticosteroids | 2.55 (2.10–3.09) | <0.001 |
| IL-6 inhibitors | 2.05 (1.49–2.83) | <0.001 |
| Viral load upon presentation | | |
| Low viral load | Reference | Reference |
| Medium viral load | 1.51 (1.24–1.83) | <0.001 |
| High viral load | 2.34 (1.96–2.80) | <0.001 |

Hospital of presentation was also included in this model (results are not shown).

Abbreviations: CI, confidence interval; IL, interleukin.

Restrictions on large gatherings began on March 12 [10]. Schools, restaurants, bars, theaters, and gyms closed on March 16, and all non-essential businesses closed on March 22 [11, 24]. Mobility patterns, as measured by mobile phone data, in NYC started to decrease in the first week of March [25]. In addition, mask wearing among the public increased in NYC during the study period; by the second week in April, 76.9% of adults in the Northeast U.S. reported using face coverings when leaving the house [26]. It is possible that a decline in viral exposure due to these practices led to declines in viral load among infected patients [27, 28]. Additional research is needed to determine the degree to which non-pharmacological interventions, such as social distancing and mask wearing, not only contribute to declines in the number of new cases, but also contribute to declines in viral load among new cases, which may in turn lead to less transmission and improved outcomes of infected patients [29].

Multiple prior studies have indicated that admission SARS-COV-2 viral load is independently associated with the risk of intubation and in-hospital mortality [5–9, 30]. Our study strengthens this evidence base by identifying an independent association between viral load and mortality in a diverse cohort of 4887 patients across multiple hospitals, even after adjusting for age, gender, common comorbidities, COVID-19 therapies, and volume of daily COVID-19 admissions. This large cohort allowed for predictive modeling to quantify the degree to which declines in viral load may have contributed to declines in mortality. We found that decreasing viral load likely had an impact on mortality, as the counterfactual model showed that lower viral loads later in the study period may have prevented 69 additional deaths (5.8% of the actual number of deaths in the study).

Our study was not designed to evaluate all factors that contributed to the decline in mortality that was observed after the peak of the pandemic in early April. In addition to viral load, other notable independent predictors of mortality in our study were age, male gender, diabetes, and volume of daily COVID-19 admissions. Changes in age and proportion of patients with diabetes did not consistently track with changes in mortality, but the proportion of male

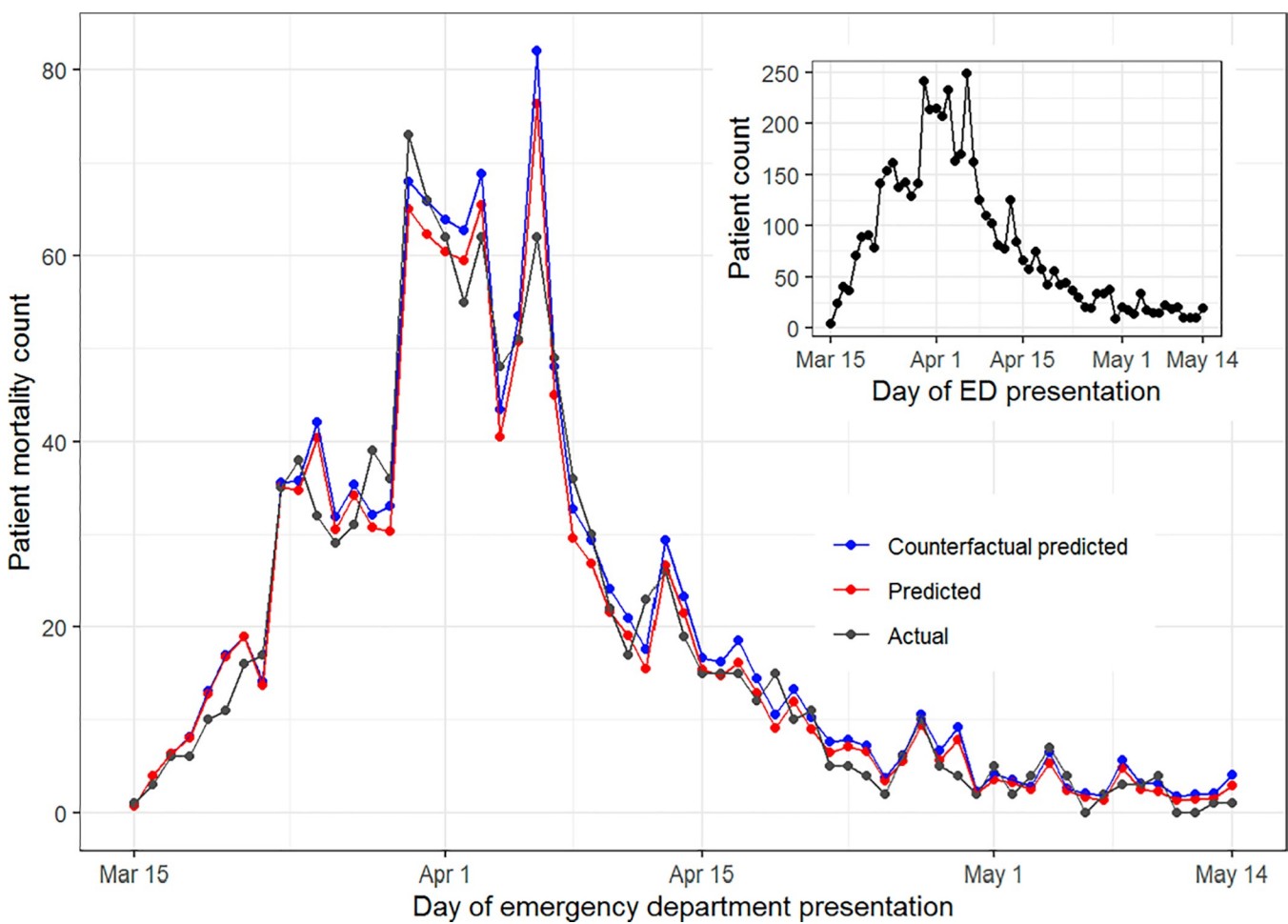

**Fig 4. Number of hospitalized patients who died during their hospitalization by day of ED presentation.** Black lines represent the actual number of deaths. Red lines represent predicted number of deaths based on the multivariable logistic regression model. Blue lines represent the predicted number of deaths if the proportions of patients with high, medium, and low viral loads had stayed the same as that observed on March 15, 2020. The Hosmer-Lemeshow test had a *P* value of 0.084 for testing general calibration and an across-time variant had a *P* value of 0.21. An additional graph in the top-right corner demonstrates the number of patients admitted with COVID-19 to study hospitals during each day of the study.

patients was highest when mortality was highest (S1 Table and Fig 3), suggesting that gender may have contributed to changes in mortality. Additionally, mortality closely paralleled the number of new admissions per day (Fig 4), suggesting that hospital overcrowding and limited resources may have played a role in the initial increase and subsequent decrease in mortality. A national study identified that in-hospital mortality decreased when the prevalence of COVID-19 in surrounding communities decreased [31], which provides additional evidence that overcrowding at hospitals may contribute to mortality.

Changes in COVID-19 therapies may also have contributed to changes in mortality. Our study predated publications from the RECOVERY trials that demonstrated a mortality benefit to dexamethasone [32] and tocilizumab [33], respectively, for patients requiring supplemental oxygen or mechanical ventilation. Thus, these medications were not consistently administered to patients in this study and there was no consistent increase in their use to attribute decreasing mortality to increased use of corticosteroids or tocilizumab. Remdesivir use increased slightly towards the end of the study, but this antiviral has not been found to decrease mortality [34]. Thus, we believe it is unlikely that antiviral or immunomodulatory interventions played a

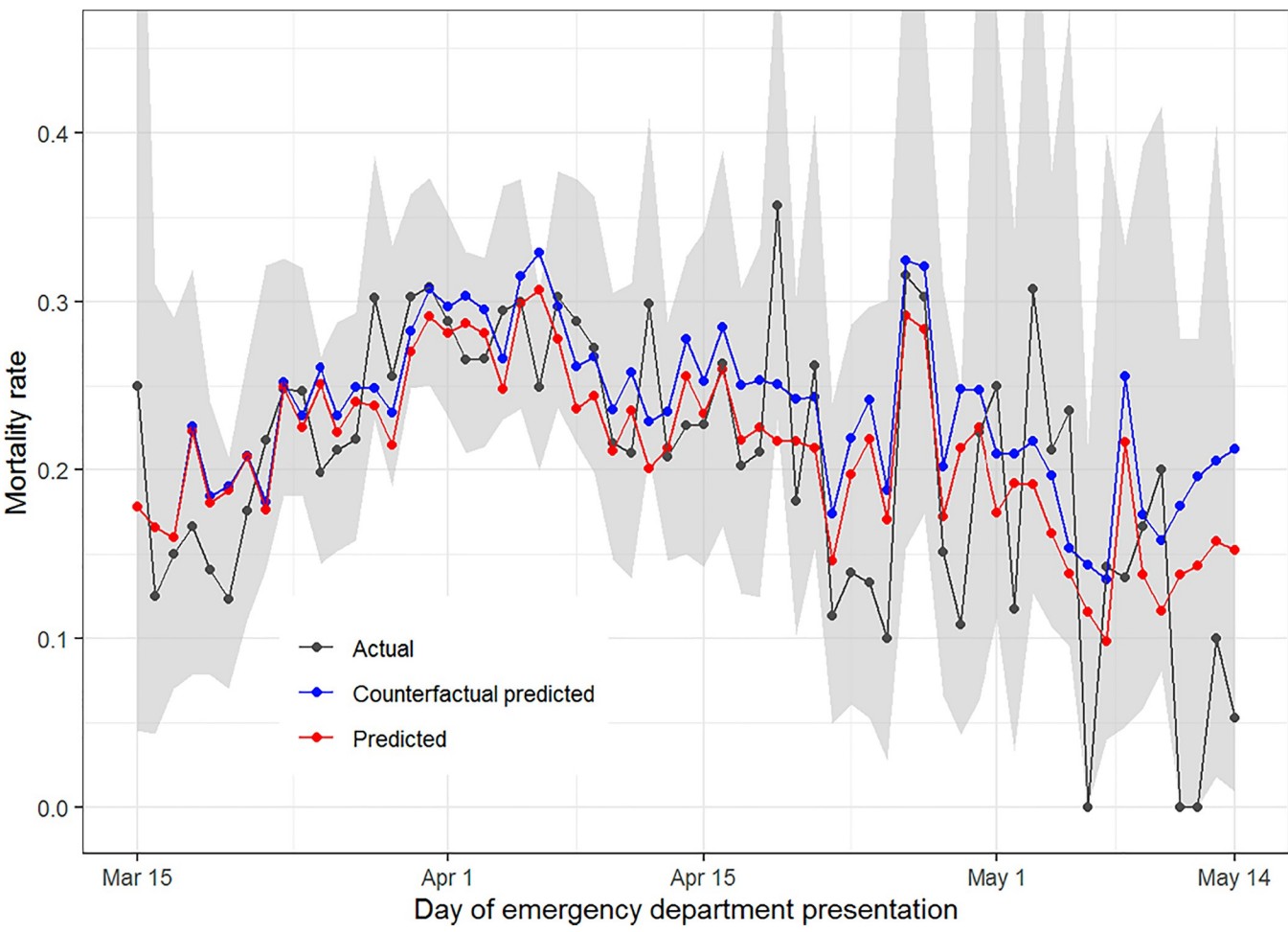

**Fig 5. Proportion of hospitalized patients who died during their hospitalization by day of ED presentation.** Black lines represent the actual number of deaths. Red lines represent predicted number of deaths based on the multivariable logistic regression model. Blue lines represent the predicted number of deaths if the proportions of patients with high, medium, and low viral loads had stayed the same as that observed on March 15, 2020.

significant role in the reduction in mortality. We were unable to analyze the use of other interventions that may have led to decreased mortality, such as the use of anti-coagulation, prone ventilation, use of high-flow oxygen devices, and general experience in the care of COVID-19 patients.

Our study has both strengths and limitations. It included several thousand patients admitted to six hospitals, but they were only from one city at the onset of the pandemic in the U.S, prior to the emergence of SARS-CoV-2 variants of concern [35] and prior to the introduction of SARS-CoV-2 vaccines [36]. We encourage other investigators to assess trends in viral load and their potential contribution to changes in mortality in other geographic regions that experienced the pandemic at different times, particularly in the setting of novel SARS-COV-2 variants, contemporary COVID-19 management strategies, and after the introduction of COVID-19 vaccines. We also did not evaluate SARS-CoV-2 viral load among outpatients because diagnostic testing was limited for these patients during the time of the study. Exploration of the clinical and epidemiologic significance of trends in viral loads among outpatients is an important area for further investigation. Finally, we used $C_T$ values as a surrogate for viral load instead of directly measuring viral load; however, the relevance of $C_T$ values as an estimate of

viral load is supported by guidance from the U.S. Food and Drug Administration that permits release of $C_T$ value results from authorized RT-PCR tests to providers [37].

Monitoring SARS-CoV-2 viral load through $C_T$ values from RT-PCR tests may be a useful tool to indicate future directions of the pandemic that may not be completely captured by merely tracking incident cases. If our findings are confirmed in other studies, the identification of increasing viral load among hospitalized patients could potentially be used as an early warning signal for an impending increase in severe cases of COVID-19 and thus inform decisions regarding closures of businesses and schools and allocation of resources for hospitals. Conversely, identification of declining viral loads among hospitalized patients could potentially be used to help guide reopenings. Furthermore, by identifying that decreases in viral load are associated with decreased mortality from COVID-19, our study establishes a foundation for further investigations to confirm whether strategies that successfully reduce exposure to SARS-CoV-2 (e.g., social distancing and mask wearing) also decrease viral load. If this is the case, then these non-pharmacological interventions would have a dual benefit, decreasing not only the risk of infection, but also the risk of mortality among infected patients.

## Supporting information

**S1 Fig. Flow diagram of patients enrolled in the study.**
(TIFF)

**S2 Fig. Plot of adjusted risk ratios of high SARS-CoV-2 viral load upon presentation to the emergency department for each study day compared to the first day of the study.** This model adjusted for age, gender, comorbidities, type of RT-PCR assay, and hospital of presentation. In the gamma regression model for $C_T$ values, the location of ED presentation (p = 0.93) and the interaction between day of ED presentation and assay type (p = 0.85) were insignificant.
(TIFF)

**S3 Fig. Plot of adjusted risk ratios of high SARS-CoV-2 viral load upon presentation to the emergency department for each study day compared to the first day of the study that includes an adjustment for duration of symptoms.** This model adjusted for the variables in S2 Fig plus duration of symptoms, including only the 50.5% of patients for whom duration of symptom data were available.
(TIFF)

**S1 Table. Characteristics of patients presenting with COVID-19 to the emergency departments of study hospitals from March 15-May 14, 2020.**
(DOC)

## Acknowledgments

We would like to acknowledge the work of our data abstraction team at Weill Cornell Medicine, Jianhua Li and Claudia Boyle from the Columbia University Department of Biomedical Informatics, the NewYork-Presbyterian Medical Technologists who performed the testing, and all of the patients and healthcare workers who cared for them.

## Author Contributions

**Conceptualization:** Michael J. Satlin, Jason Zucker, Benjamin R. Baer, Mangala Rajan, Nathaniel Hupert, Luis M. Schang, Laura C. Pinheiro, Magdalena E. Sobieszczyk, Lars F. Westblade, Parag Goyal, Jorge L. Sepulveda, Monika M. Safford.

**Data curation:** Michael J. Satlin, Jason Zucker, Laura C. Pinheiro, Magdalena E. Sobieszczyk, Lars F. Westblade, Parag Goyal, Jorge L. Sepulveda, Monika M. Safford.

**Formal analysis:** Michael J. Satlin, Jason Zucker, Benjamin R. Baer, Mangala Rajan, Nathaniel Hupert, Luis M. Schang, Laura C. Pinheiro, Yanhan Shen, Lars F. Westblade, Parag Goyal, Martin T. Wells, Jorge L. Sepulveda, Monika M. Safford.

**Investigation:** Michael J. Satlin, Jason Zucker, Benjamin R. Baer, Mangala Rajan, Luis M. Schang, Laura C. Pinheiro, Lars F. Westblade, Parag Goyal, Jorge L. Sepulveda, Monika M. Safford.

**Methodology:** Michael J. Satlin, Jason Zucker, Benjamin R. Baer, Mangala Rajan, Nathaniel Hupert, Luis M. Schang, Laura C. Pinheiro, Yanhan Shen, Lars F. Westblade, Parag Goyal, Jorge L. Sepulveda, Monika M. Safford.

**Project administration:** Michael J. Satlin, Jason Zucker, Laura C. Pinheiro, Parag Goyal, Monika M. Safford.

**Resources:** Michael J. Satlin, Jason Zucker, Benjamin R. Baer, Mangala Rajan, Laura C. Pinheiro, Parag Goyal, Monika M. Safford.

**Software:** Michael J. Satlin, Jason Zucker, Benjamin R. Baer, Mangala Rajan, Yanhan Shen, Parag Goyal, Jorge L. Sepulveda.

**Supervision:** Michael J. Satlin, Jason Zucker, Benjamin R. Baer, Mangala Rajan, Lars F. Westblade, Parag Goyal, Martin T. Wells, Monika M. Safford.

**Validation:** Michael J. Satlin, Jason Zucker, Benjamin R. Baer, Mangala Rajan, Lars F. Westblade.

**Visualization:** Michael J. Satlin, Jason Zucker, Benjamin R. Baer.

**Writing – original draft:** Jason Zucker, Benjamin R. Baer, Mangala Rajan, Laura C. Pinheiro, Monika M. Safford.

**Writing – review & editing:** Jason Zucker, Benjamin R. Baer, Mangala Rajan, Nathaniel Hupert, Luis M. Schang, Laura C. Pinheiro, Yanhan Shen, Magdalena E. Sobieszczyk, Lars F. Westblade, Parag Goyal, Martin T. Wells, Jorge L. Sepulveda, Monika M. Safford.

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
