## [Decision Letter · Decision Letter 0]

19 Aug 2021

PONE-D-21-21421

Changes in SARS-CoV-2 viral load and mortality during the initial wave of the pandemic in New York City

PLOS ONE

Dear Dr. Satlin,

Thank you for submitting your manuscript to PLOS ONE. After careful consideration, we feel that it has merit but does not fully meet PLOS ONE’s publication criteria as it currently stands. Therefore, we invite you to submit a revised version of the manuscript that addresses the points raised during the review process.

Please revise accordingly.

We look forward to receiving your revised manuscript.

Kind regards,

Academic Editor

PLOS ONE

Journal Requirements:

Reviewers' comments:

Reviewer's Responses to Questions

**Comments to the Author**

1. Is the manuscript technically sound, and do the data support the conclusions?

Reviewer #1: Yes

Reviewer #2: Yes

2. Has the statistical analysis been performed appropriately and rigorously? 

Reviewer #1: Yes

Reviewer #2: Yes

3. Have the authors made all data underlying the findings in their manuscript fully available?

Reviewer #1: Yes

Reviewer #2: Yes

4. Is the manuscript presented in an intelligible fashion and written in standard English?

Reviewer #1: No

Reviewer #2: Yes

5. Review Comments to the Author

Reviewer #1: Author s of this manuscript tried to describe the effects of public health intervention such as social distancing, mask wearing etc. to viral load and mortality in SARS-Cov2 infection in New city during the initial stage of pandemic. They analyzed all most 7,000 patients in different hospital in New York city. They find out the incfease of Ct values and decrease of mortality during this investigations after application of public health intervention. So, by setting of publich health intervention, the viral load was declined in the hospitalized patients and correlated with the

decrease of mortality.

Generally, this study is well done and showed the effects of public health interventions to viral load and mortality of SARS-CoV2 infected patients. So, this manuscript will be very useful in the current pandemic. Howeverm there are somethings to be addressed.

o There are some typo errors and authors should use scientific words.

o If authors add some information related with other respiratory infections, value of this manuscript will be improved.

Reviewer #2: Thanks for sending the manuscript for review titled "Changes in SARS-CoV-2 viral load and mortality during the initial wave of the pandemic in New York City." The manuscript highlighted the importance of public health interventions which is already proven effective. The draft is already in a very good shape. I have two minor suggestions:

1. It would be great if authors could add one sentence about the reason of selecting those six hospitals in methods section.

2. In addition, authors could mention the exact start date of implementing non-pharmacological public health interventions to reduce interpersonal interactions and aerosol exposures in NYC. It could strengthen the argument in the discussion section.

Thanks.

6. PLOS authors have the option to publish the peer review history of their article (what does this mean?). If published, this will include your full peer review and any attached files.

Reviewer #1: No

Reviewer #2: **Yes: **Mahbub-Ul Alam

---

## [Author Response · Author response to Decision Letter 0]

27 Aug 2021

Journal Requirements:

Response: We have formatted the manuscript to meet PLOS ONE’s style requirements

Response: An ethics statement was added to the Methods section, including the IRBs who approved the study and that the study was conducted with a waiver of informed consent (lines 104-106, track changes version).

Response: All references have been reviewed and edited for accuracy. We corrected the publication year for reference 18. No cited papers have been retracted. 

Reviewer #1:

1) There are some typo errors and authors should use scientific words.

Response: We have reviewed the manuscript for typo errors and have ensured that scientific words were used throughout the manuscript. 

2) If authors add some information related with other respiratory infections, value of this manuscript will be improved

Response: Unfortunately, we do not have data on other respiratory viruses that occurred at study hospitals during the study period. In fact, many of the hospitals did not test for other respiratory viruses during the initial wave of the pandemic to maintain the resources needed for the incredibly high testing volume for SARS-CoV-2.

Reviewer #2:

1) It would be great if authors could add one sentence about the reason of selecting those six hospitals in methods section.

Response: We added to a sentence in the Methods section that outlines why these hospitals were selected for the study (lines 92-96, track changes version). 

2) In addition, authors could mention the exact start date of implementing non-pharmacological public health interventions to reduce interpersonal interactions and aerosol exposures in NYC. It could strengthen the argument in the discussion section.

Response: We added two sentences to the Discussion that outline the exact dates that non-pharmacological interventions were initiated in New York City (lines 303-305, track changes version).

---

## [Decision Letter · Decision Letter 1]

15 Sep 2021

Changes in SARS-CoV-2 viral load and mortality during the initial wave of the pandemic in New York City

PONE-D-21-21421R1

Dear Dr. Satlin,

We’re pleased to inform you that your manuscript has been judged scientifically suitable for publication and will be formally accepted for publication once it meets all outstanding technical requirements.

Kind regards,

Academic Editor

PLOS ONE

Additional Editor Comments (optional):

Reviewers' comments:

Reviewer's Responses to Questions

**Comments to the Author**

1. If the authors have adequately addressed your comments raised in a previous round of review and you feel that this manuscript is now acceptable for publication, you may indicate that here to bypass the “Comments to the Author” section, enter your conflict of interest statement in the “Confidential to Editor” section, and submit your "Accept" recommendation.

Reviewer #1: All comments have been addressed

Reviewer #2: All comments have been addressed

2. Is the manuscript technically sound, and do the data support the conclusions?

Reviewer #1: Yes

Reviewer #2: Yes

3. Has the statistical analysis been performed appropriately and rigorously? 

Reviewer #1: Yes

Reviewer #2: Yes

4. Have the authors made all data underlying the findings in their manuscript fully available?

Reviewer #1: Yes

Reviewer #2: Yes

5. Is the manuscript presented in an intelligible fashion and written in standard English?

Reviewer #1: Yes

Reviewer #2: Yes

6. Review Comments to the Author

Reviewer #1: Authors very well addressed all questions raised from me. So, this revision is suitable to publish in this journal.

Reviewer #2: Thanks for the revision. Authors have addressed all comments by reviewers. I do not have additional comments.

7. PLOS authors have the option to publish the peer review history of their article (what does this mean?). If published, this will include your full peer review and any attached files.

Reviewer #1: No

Reviewer #2: **Yes: **Mahbub-Ul Alam

---

## [Editor Report · Acceptance letter]

10 Nov 2021

PONE-D-21-21421R1 

Changes in SARS-CoV-2 viral load and mortality during the initial wave of the pandemic in New York City 

Dear Dr. Satlin:

I'm pleased to inform you that your manuscript has been deemed suitable for publication in PLOS ONE. Congratulations! Your manuscript is now with our production department. 

Kind regards, 

on behalf of

Dr. Robert Jeenchen Chen 

Academic Editor

PLOS ONE